# DROID: Learning from Offline Heterogeneous Demonstrations via Reward-Policy Distillation

**Sravan Jayanthi**[*,1], **Letian Chen**[*,†,1], **Nadya Balabanska**[2], **Van Duong**[2], **Erik Scarlatescu**[1],
**Ezra Ameperosa**[1], **Zulfiqar Zaidi**[1], **Daniel Martin**[1], **Taylor Del Matto**[1], **Masahiro Ono**[2],
**Matthew Gombolay**[1]

[1]School of Interactive Computing, Georgia Institute of Technology
[2]Jet Propulsion Laboratory, California Institute of Technology
[*]Equal Contribution, [†]Corresponding Author: `letian.chen@gatech.edu`

**Abstract:** Offline Learning from Demonstrations (OLfD) is valuable in domains where trial-and-error learning is infeasible or specifying a cost function is difficult, such as robotic surgery, autonomous driving, and path-finding for NASA's Mars rovers. However, two key problems remain challenging in OLfD: 1) *heterogeneity*: demonstration data can be generated with diverse preferences and strategies, and 2) *generalizability*: the learned policy and reward must perform well in unseen test settings beyond the limited training regime. To overcome these challenges, we propose Dual Reward and policy Offline Inverse Distillation (DROID) that leverages diversity to improve generalization performance by decomposing common-task and individual-specific strategies and distilling knowledge in both the reward and policy spaces. We ground DROID in a novel and uniquely challenging Mars rover path-planning problem for NASA's Mars Curiosity Rover. We curate a novel dataset along 154 Sols (Martian days) and conduct a novel, empirical investigation to characterize heterogeneity in the dataset. We find DROID outperforms prior SOTA OLfD techniques, leading to a 21% improvement in modeling expert behaviors and 90% closer to the task objective of reaching the final destination. We also benchmark DROID on the OpenAI Gym Cartpole and Lunar Lander environments and find DROID achieves 23% (significantly) better performance modeling unseen holdout heterogeneous demonstrations.

**Keywords:** Learning from Heterogeneous Demonstration, Network Distillation, Offline Imitation Learning

## 1 Introduction

Deep Reinforcement Learning (Deep RL) has achieved success in generating high-performance continuous control behaviors but requires a high-fidelity simulator or reward-annotated dataset [1, 2, 3, 4, 5, 6, 7, 8]. A counterexample is the Mars Path-Planning (MPP) problem where one must construct a path through a series of waypoints for the Mars rover to traverse towards a destination without an explicit notion of reward or simulator. Not only is this challenging due to the chaotic terrain but also because of factors such as physical limitations of the rover's capabilities and unobservable terrain information [9, 10]. Expert human Rover Planners (RPs) at NASA design paths under time and safety constraints based on their expertise crafted over years of experience – knowledge that has yet to be codified [9]. Efforts have been made to automate the process with symbolic and connectionist (e.g., Deep RL) approaches [11, 12, 13]. However, these methods do not match the human RPs' success because it is difficult to codify experts' knowledge into a cost function [13, 14], and a large homogeneous dataset is usually required to learn robust policies [9]. Such challenges not only exist in the MPP problem but are prevalent in other robotic applications such as surgery, search and rescue, self-driving, and elderly care [15, 16, 17, 18, 19, 20, 21].

7th Conference on Robot Learning (CoRL 2023), Atlanta, USA.

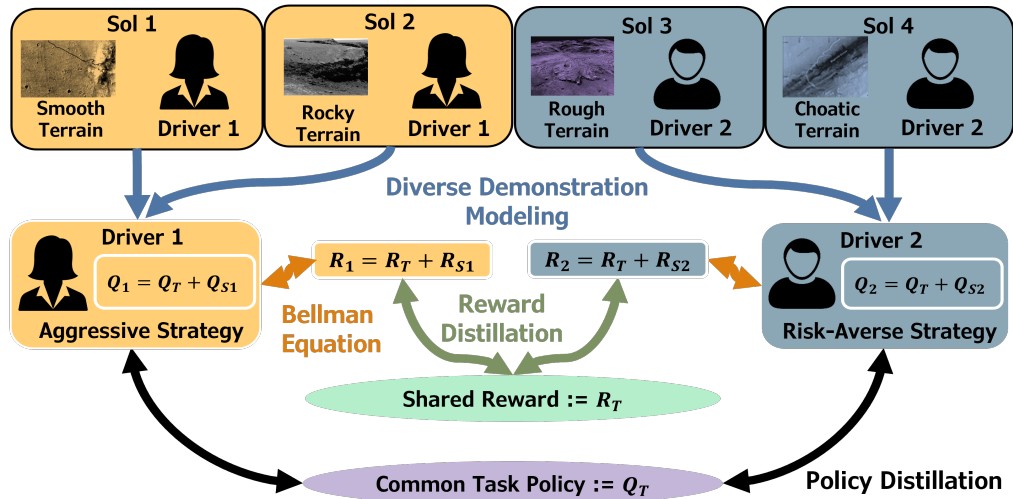

Figure 1: This figure shows how DROID infers (blue arrows) the underlying expert strategies given the MPP dataset with varying preferences (e.g., Aggressive vs. Risk-Averse) across heterogeneous states (Smooth to Chaotic Terrains). DROID performs knowledge distillation to a common task policy (black arrows) shared by all demonstrators, infers a reward that encodes each expert's latent preference (orange arrows), and identifies the shared reward across demonstrators (green arrows).

Learning from Demonstration (LfD) is a promising paradigm to address this challenge: LfD methods learn by having users demonstrate the desired behavior on the robot, removing the need for cost function specification [22]. However, most LfD approaches are limited by the need for many environment interactions [23, 24, 25]. In robotic applications, the exploratory environment interactions could be costly (e.g., damaged or lost rovers), unethical (e.g., in surgery), or unsafe. Appropriately, Offline LfD (OLfD) has been proposed as a framework that allows for training a robot policy solely from pre-recorded demonstrations with no assumption about a viable simulator [26].

OLfD relaxes the requirement for a reward function and a simulator, it however faces several algorithmic challenges that limit its full potential [26, 27]. First, a key challenge for OLfD is heterogeneity within the demonstration set. Each expert has individual preferences stemming from varying cognitive biases [28, 29] or different latent goals [30, 31] for accomplishing a given task. If the LfD algorithm assumes homogeneity about heterogeneous data, the robot may fail to infer the expert intention [28, 32] and the learned policy may perform poorly [28, 33]. On the other hand, learning each heterogeneous behavior separately is data-inefficient and prone to overfitting. The second critical challenge is learning from a limited dataset [34]. The limited data makes it difficult for the learned policy and reward functions to capture the user's latent intentions and to generalize beyond the demonstrated setting [15, 35, 36, 37].

In this paper, we propose a novel OLfD approach, **D**ual **R**eward and Policy **O**ffline **I**nverse **D**istillation (DROID), that simultaneously distills a common *task policy* and *reward* from diverse demonstrators, while modeling individual preferences along both *strategy-specific policies* and *rewards*. This approach allows us to extract an unbiased task objective while understanding various styles of accomplishing the task (Figure 1). Our contributions are three-fold:

1. We curate a novel dataset with RP-designed Mars Curiosity Rover paths for 154 Sols (Martian days) covering various terrains on Mars. We conduct a novel, empirical investigation to characterize heterogeneity in the dataset, motivating the need for a OLfD approach robust to heterogeneity.

2. We propose DROID, a framework that simultaneously distills knowledge through the learned policy and reward. We also introduce two improvements (Augmented Regularization & Reward Maximization) to the underlying IRL algorithm [15] to improve generalization performance.

3. We show DROID achieves 29% and 17% better modeling performance (measuring the distance between expert demonstration and the generated trajectory) than previous SOTA in Cartpole and Lunar Lander, respectively. On the MPP problem, we also find DROID outperforms SOTA and gets 90% closer to the goal point (an important objective in the Mars path planning domain).

## 2 Related Work

In this section, we discuss related works in offline LfD, reward and policy distillation, and the Mars Path Planning problem. There has been extensive work for LfD in robotics and learning from heterogeneous demonstrations [30, 29, 33, 38, 39, 40, 41, 42, 43] but our focus is offline learning from diverse, limited demonstrations, an unexplored problem setting.

**Offline LfD –** Few previous OLfD approaches model demonstration heterogeneity [44, 45, 46, 47, 48], which could cause the learned reward function and policy to fail at generalizing beyond the original demonstrated setting [35, 36] and capturing the individual preference of the expert [37]. Some work in OLfD attempts to tackle multimodal expert behaviors by increasing model capacity through Diffusion [49], BeT (Behavior Transformers) [50], and representation learning [51], but each fails to overcome the fundamentally mode-seeking behavior of LfD. DROID's explicit policy and reward decomposition is critical to success in modeling heterogeneity.

**Reward and Policy Distillation –** Several frameworks consider commonalities among reward functions across heterogeneous demonstrations [29, 38, 41]. However, these methods rely on online interactions with the environment, which is infeasible in many robotic domains. Policy distillation has been studied to improve policy transfer performance [52, 53]. However, DROID is the first to study simultaneous reward and policy distillation, particularly in the challenging setting of OLfD.

**Mars Path Planning –** For NASA, it requires much human effort to plan paths for the Mars Curiosity Rover [9]. Current approaches such as Autonomous Navigation (AutoNav) [54] do not consider all hazards that humans deem dangerous to the rover [55]. Hedrick et al. [56] proposes efficient Martian path planning, and Rover-IRL [57] learns a cost function from demonstration, but both fail to plan under missing/occluded terrain maps, which is a key obstacle in the Mars domain [9]. Our algorithm, DROID, instead directly learns from how previous RPs drive in the midst of occluded info and has the ability to model an RP's behavior when planning along unknown parts of the terrain.

## 3 Preliminaries

In this section, we introduce preliminaries on Markov Decision Processes (MDP), Offline Learning from Demonstration (OLfD), and Multi-Strategy Reward Distillation (MSRD).

**Markov Decision Process –** A MDP, $M$, is a 6-tuple, $\langle \mathbb{S}, \mathbb{A}, R, T, \gamma, \rho_0 \rangle$. $\mathbb{S}$ and $\mathbb{A}$ correspond to the state and action spaces, $R(s, a)$ the reward, and $T(s'|s, a)$ the transition probability for state $s'$ after performing action $a$ in state $s$. $\gamma \in (0, 1)$ is the discount factor and $\rho_0$ denotes the initial state distribution. The policy $\pi(a|s)$ represents the probability of choosing action $a$ in state $s$. The Q-value is defined as $Q_R^\pi(s, a) = \mathbb{E}_{\pi,T} \left[ \sum_{t=0}^\infty \gamma^t R(s_t, a_t) | s_0 = s, a_0 = a \right]$, denoting the expected discounted cumulative return following $\pi$ in the MDP under reward $R$.

**Offline Learning from Demonstration –** IRL considers an MDP sans reward function (MDP\R) and infers the reward function $R$ based on a set of demonstration trajectories $\mathcal{D} = \{\tau_1, \tau_2, \cdots, \tau_N\}$, where $N$ is the number of demonstrations. Our method leverages Approximate Variational Reward Imitation Learning (AVRIL) [58] as the underlying IRL approach. AVRIL considers a distribution over the reward function and approximates the posterior, $p(R)$, with a variational distribution, $q_\phi(R)$. AVRIL introduces a second variational approximation for $Q_R^{\pi_E}(s, a)$ with $Q_\theta(s, a)$ and ensures the variational reward distribution, $q_\phi(R)$, is consistent with the variational Q function, $Q_\theta(s, a)$, by Bellman equation. We show the final objective function of AVRIL in Equation 1.

$$L_{\text{AVRIL}} = \sum_{(s,a,s',a') \in \mathcal{D}} \log \left[ \frac{\exp(\beta Q_\theta(s, a))}{\sum_{b \in A} \exp(\beta Q_\theta(s, b))} \right] - D_{\text{KL}}(q_\phi(R) \| p(R)) + \lambda \log q_\phi(Q_\theta(s, a) - \gamma Q_\theta(s', a')) \quad (1)$$

**Multi-Strategy Reward Distillation –** We propose a general reward distillation framework based on a previous online IRL technique, MSRD [38]. MSRD decomposes the per-strategy reward, $R_i$, for strategy $i$, as a linear combination of a common task reward and a strategy-only reward with neural network parameters $\phi_{\text{Task}}$ and $\phi_{\text{S}-i}$: $R_i = R_{\phi_{\text{Task}}} + R_{\phi_{\text{S}-i}}$. MSRD leverages a regularization loss to distill common knowledge into $\phi_{\text{Task}}$ and retain personalized information in $\phi_{\text{S}-i}$.

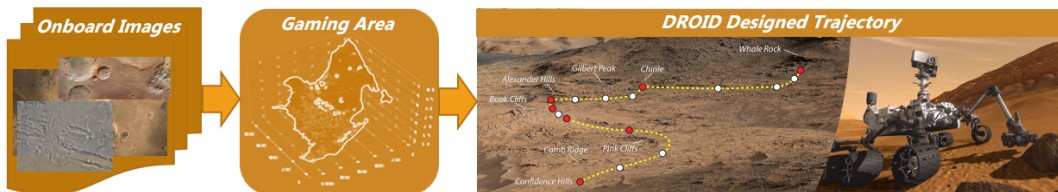

Figure 2: This figure shows the dataset curation process for the MPP problem. We unify the height maps created by onboard cameras into a single "gaming area" (middle figure) and then plan the driving path based on features calculated on the gaming area height map.

## 4   Mars Curiosity Rover Path Planning Problem

In this section, we introduce the Mars Path Planning (MPP) problem and how we 1) curate the dataset, 2) construct an MDP for OLfD, 3) analyze heterogeneity present across RPs.

**Dataset Curation** The raw data consists of height maps created by photos captured by Curiosity across multiple Sols (Martian days). The multi-resolution height maps are processed into a single 64x64 "gaming area" by interpolation of overlapping height maps and scaling along each axis (Figure 2). The processed gaming area is then used to calculate nine features identified by RPs: (1) distance to the goal point, (2) unknown data percentage, (3) average roughness, (4) maximum roughness, (5) average pitch, (6) average roll, (7) maximum pitch, (8) maximum roll, and (9) turning angle. More details for the features are available in the Supplementary.

**MDP Problem Setup** We create a novel formulation to convert the MPP problem into an MDP: a state contains the terrain information of the Sol and the current and target locations for the path planning. The action space, $A$, consists of all possible next waypoints in the gaming area. The reward function is constructed as a function of features associated with the path specified by $a$ on the terrain $s$: $R(s, a) = f(\psi(s, a))$ where $\psi : (S \times A) \to \mathbb{R}^9$ is the path feature mapping.

**Analysis of Heterogeneity** We perform a PER-MANOVA test with $\alpha = 0.05$ and the Holm method for the correction of multi-tests with the 37 different RPs to answer whether different strategies exist among drivers in the feature space. The test shows significant differences along seven pairs of RPs, particularly along the paths designed by RP 26 with respect to 5 other RPs, as shown in Figure 3 with TSNE [59] to reduce features into two dimensions (further explanation provided in the supplementary). This result shows heterogeneity in the RP-generated paths from differences in terrains of the Sols and the expert RP strategies, motivating the need for offline learning from *heterogeneous* demonstration approaches.

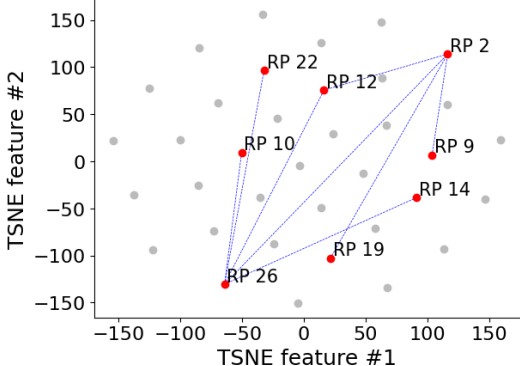

Figure 3: This figure shows heterogeneity in MPP dataset. Comparisons with $p < .05$ are represented by connecting lines. RP IDs are labeled and marked as red.

## 5   Methods

The challenges of heterogeneity and limited data are prevalent for OLfD in many robotic applications, particularly in the MPP problem as shown in Section 4. To overcome these challenges, we propose our algorithm, **DROID**, for learning high-performing policies by distilling common information across heterogeneous demonstrations in both the policy space and the reward function space.

### 5.1   Reward Distillation

We propose a general reward distillation approach for OLfD. We model the reward distributions as mean-field Gaussian distributions partitioned on each state-action pair and let the reward neural networks output the mean and standard deviation of the Gaussian distributions. The advantage of Gaussian-distribution models for task reward ($\mathfrak{R}_{\text{Task}}(s, a) \sim q_{\phi_{\text{Task}}}(R) = \mathcal{N}(\mu_{\text{Task}}(s, a), \sigma^2_{\text{Task}}(s, a))$)

and strategy rewards ($\mathfrak{R}_{\text{S}-i}(s,a) \sim q_{\phi_{\text{S}-i}}(R) = \mathcal{N}(\mu_{\text{S}-i}(s,a), \sigma^2_{\text{S}-i}(s,a)))$ is that the summation of two Gaussian distributions is still Gaussian distribution, as shown in Equation 2, where $\mathfrak{R}_i$ denotes the random variable for the reward distribution of strategy $i$.

$$\mathfrak{R}_i(s,a) = \mathfrak{R}_{\text{Task}}(s,a) + \mathfrak{R}_{\text{S}-i}(s,a) \sim \mathcal{N}(\mu_{\text{Task}}(s,a) + \mu_{\text{S}-i}(s,a), \sigma^2_{\text{Task}}(s,a) + \sigma^2_{\text{S}-i}(s,a)) \quad (2)$$

Assuming the number of strategies, $M$, and the strategy label, $c_\tau$ for $\tau$, is known a priori, we can perform reward distillation on the strategy reward, as shown in Equation 3.

$$L_{\text{RD}}(\{\phi_{\text{S}-i}\}_{i=1}^M; \mathcal{D}) = \mathbb{E}_{(\tau, c_\tau) \in D} [||\mu_{\text{S}-c_\tau}(s,a)||] \quad (3)$$

Intuitively, $L_{\text{RD}}$ pushes the strategy reward to output 0 and therefore encourages common knowledge to flow to the shared task reward and each individual strategy reward only captures preferences. The MSRD reward distillation formulation is a special case where the reward distribution for each strategy collapses to a Dirac delta function.

## 5.2 Policy Distillation

We propose DROID to leverage commonalities in both reward and policy spaces. As policies are implicitly defined via the $Q$ function in AVRIL by $\pi(a|s) = \arg\max_{a \in A} Q(s,a)$, we construct the Q function for each strategy as a combination of task Q function and strategy Q function: $Q_i = Q_{\theta_{\text{Task}}} + Q_{\theta_{\text{S}-i}}$. As such, we propose to regularize the output of the strategy Q values, $Q_{\text{S}-i}$, as in Equation 4, to encourage common information to be distilled into the task Q-value, $Q_{\text{Task}}$.

$$L_{\text{PD}}(\{\theta_{\text{S}-i}\}_{i=1}^M; \mathcal{D}) = \mathbb{E}_{(\tau, c_\tau) \in D} [||Q_{\text{S}-c_\tau}(s,a)||] \quad (4)$$

DROID's explicit knowledge distillation across diverse policies aids in improving generalization performance as the shared policy benefits from modeling all demonstrations in the offline dataset.

## 5.3 Enhancing DROID for Offline LfD

We present two enhancements to construct the inductive bias useful for learning more accurate rewards and better-performing policies in OLfD.

**Improvement 1: Augmenting Dataset for Regularization.** We introduce an augmented dataset, $\mathcal{D}' = \{(s,b)|s \in \mathcal{D}, b \in \mathbb{A}\}$, for regularizing $q_\phi(R)$ to a prior $p(R)$ compared with AVRIL's KL divergence regularization only within the demonstration (Equation 1). By extending the operation of $D_{\text{KL}}$ to be on the entire action space, we are encouraging a conservative estimate of the reward for any action that is not taken by demonstrators, following the pessimistic principle in OLfD [60].

$$L_{\text{KL}}^+(\phi; \mathcal{D}) = \sum_{s \in \mathcal{D}, a \in \mathbb{A}} D_{\text{KL}}(q_\phi(R(s,a))||p(R(s,a))) \quad (5)$$

Intuitively, Equation 5 could be viewed as a data augmentation technique to regularize reward learning across the entire action space. In contrast, AVRIL's variational lower-bound only regularizes $L_{\text{KL}}(\phi) = \sum_{(s,a) \in \mathcal{D}} D_{\text{KL}}(q_\phi(R(s,a)||p(R(s,a)))$, on $(s,a)$ samples from $\mathcal{D}$. We provide a lemma to describe the effect of this augmented regularization in the supplementary.

**Improvement 2: Reward Maximization.** The second improvement we propose is to maximize the reward given to the demonstrated action, as shown in Equation 6.

$$L_{\text{max-action-reward}}(\phi; \mathcal{D}) = - \sum_{(s,a) \in \mathcal{D}} r(s,a) \quad \text{where } r(s,a) \sim q_\phi(R(s,a)) \quad (6)$$

In AVRIL, the reward function learning relies on the two-stage process of 1) Q function learning (first-term of Equation 1) and then 2) reward learning by compatibility (third-term of Equation 1). Our proposed $L_{\text{max-action}}$ instead directly encourages high reward for demonstrated state-action pairs, allowing the reward learning to be faster without reliance on a successful Q function learning. Combining these two enhancements and distillations on reward and policy, we summarize the loss function for DROID in Equation 7. We also provide more details regarding the two improvements and a pseudocode for DROID in the supplementary.

$$L = \sum_{(s,a) \in \mathcal{D}} \log \frac{\exp \beta Q_\theta(s,a)}{\sum_{b \in A} \exp(\beta Q_\theta(s,b))} - \sum_{s \in \mathcal{D}, a \in \mathbb{A}} D_{\text{KL}}(q_\phi(R(s,a))||p(R(s,a))) + L_{\text{max-action}}(\phi; \mathcal{D})$$
$$+ \sum_{(s,a,s',a') \in \mathcal{D}} \lambda \log q_\phi(Q_\theta(s,a) - \gamma Q_\theta(s',a')) + L_{\text{RD}}(\{\phi_{\text{S}-i}\}_{i=1}^M; \mathcal{D}) + L_{\text{PD}}(\{\theta_{\text{S}-i}\}_{i=1}^M; \mathcal{D})$$
$$(7)$$

# 6 Results

In this section, we show that DROID achieves strong performance in two OpenAI Gym environments (CartPole and LunarLander) [61] and the more difficult Mars Path Planning problem compared to prior works (Section 6.1-6.2) and DROID's own ablations (Section 6.3). We focus our analysis on three questions that address the 3 challenges in Offline LfD: heterogeneity, policy generalizability to unseen task settings, and reward transferrability to downstream tasks.

**Q1: Diverse Demonstration Modeling** – How well does DROID perform at modeling different preferences from heterogeneous demonstrations in Offline LfD?

**Q2: Policy Generalizability** – How well can the learned policies perform in an unseen holdout test dataset (e.g., modeling unseen demonstrations in CartPole, planning unseen terrains in MPP)?

**Q3: Reward Generalizability** – How successful are the learned rewards in encoding experts' latent objectives and inducing high-performing downstream policies on unseen test settings?

For the benchmark experiments, we compare DROID against a collection of OLfD baselines: a) Behavior Cloning (BC) Batch: a single BC model across the dataset [44, 45], b) BC Single, which trains a BC model for each demonstrator, c) Diffusion model [49] which is a generative modeling technique that leverages denoising to model multimodal demonstrations, d) Behavior Transformers (BeT) [50], which models next actions conditioned on the sequence observations, e) AVRIL Batch, a single AVRIL model on all data, f) AVRIL Batch XL, which increases model capacity fourfold to better model the heterogeneous dataset, g) AVRIL State Representation Learning (SRL) [51] which implicitly models multimodal data by inducing a representation space trained by predicting the corresponding action, h) AVRIL Single, training separate models for each expert and i) MSRD-Offline: MSRD with reward distillation adapted from DROID.

## 6.1 Cartpole and LunarLander

We evaluate DROID against baselines on four metrics: Frechet Distance [62], KL Divergence [63], Undirected Hausdorff Distance [64], and Average Log Likelihood. We train each method on a dataset of 60 trajectories from 20 distinct strategies generated by jointly optimizing an environment reward and a diversity reward from DIAYN [65]. Diverse CartPole strategies include swinging to different ends of the track and oscillating at varying periodicity while diverse Lunar Lander (LL) strategies include different landing attack angles and touchdown techniques. More experiment details and videos of demonstrations and learned policies are provided in the supplementary.

**Q1: Diverse Demonstration Modeling.** Table 1 summarizes the results of modeling and imitation on the training demonstrations. We find DROID performs significantly ($p < .05$) better on the Undirected Hausdorff (19% in Cartpole, 20% in LL) and Frechet Distance (32% in Cartpole) compared to the best baselines, showing that DROID models heterogeneous behaviors better and minimizes deviation between the learned policy's behavior and the expert demonstrations.

**Q2: Policy Generalizability.** We study how well each method's policy performs on unseen demonstrations. Results in Table 1 section "Policy Generalizability" show DROID significantly outperforms ($p < .05$) baselines along Frechet and Undirected Hausdorff. Especially, DROID's performance gain over MSRD-Offline being larger on generalization than imitation shows the policy distillation in DROID is essential to learn a generalizable policy.

**Q3: Reward Generalizability.** As a further analysis of generalization performance, we study how successful the learned reward functions are at inducing high-performing policies. We train offline RL policies with CQL [66] and compare performances on the holdout test set. The results in Table 1 section "Reward Generalizability" show DROID achieves significantly better ($p < .05$) performance on log-likelihood (underperforms on Frechet Distance and Undirected Hausdorff), showing DROID's reward function can induce a similarly well-performing policy.

## 6.2 Mars Path Planning

With the success of DROID on Cartpole, we further test it against benchmarks on the more challenging MPP problem, where RPs optimize for a complex objective considering goal locations, strategies, and safety constraints. Since there is no clear ground truth reward in the MPP problem, we study the performance along four metrics: Distance from (expert) Waypoint, Final Distance (from

Table 1: This table shows performance comparisons and significance of DROID and baselines in Cartpole (left) and Lunar Lander (right). Bold denotes the best-performing model for the metric. * denotes significance of $p < .05$ against the second-best model.

**CartPole**

| Benchmark Method | KL Divergence | Frechet Distance | Undirected Hausdorff | Log Likelihood |
|---|---|---|---|---|
| **Diverse Demonstration Modeling ($n = 40$)** | | | | |
| BC Batch | 10.046 | 1.192 | 0.969 | -25.599 |
| BC Single | 9.440 | 1.176 | 0.923 | -24.729 |
| Diffusion | 13.687 | 2.922 | 2.867 | -165.517 |
| BeT | 13.755 | 2.901 | 2.836 | -138.629 |
| AVRIL Batch | 7.608 | 0.933 | 0.729 | -48.113 |
| AVRIL Batch XL | 8.840 | 1.023 | 0.775 | -45.991 |
| AVRIL SRL | 11.335 | 1.395 | 1.069 | **-28.713** |
| AVRIL Single | 10.051 | 1.294 | 0.895 | -48.910 |
| MSRD-Offline | 7.479 | 0.621 | 0.476 | -40.453 |
| DROID (ours) | **6.047** | **0.425*** | **0.261*** | -37.948 |
| **Policy Generalizability ($n = 20$)** | | | | |
| BC Batch | 10.792 | 1.237 | 1.026 | -33.079 |
| BC Single | 9.330 | 1.111 | 0.881 | -32.018 |
| Diffusion | 13.825 | 3.050 | 2.911 | -164.980 |
| BeT | 13.864 | 2.959 | 2.853 | -138.629 |
| AVRIL Batch | 8.367 | 1.004 | 0.786 | -52.843 |
| AVRIL Batch XL | 7.878 | 0.950 | 0.738 | -50.698 |
| AVRIL SRL | 11.320 | 1.449 | 1.056 | **-34.710** |
| AVRIL Single | 7.960 | 1.006 | 0.717 | -54.023 |
| MSRD-Offline | 7.582 | 0.584 | 0.458 | -44.173 |
| DROID (ours) | **5.271** | **0.412*** | **0.207*** | -38.057 |
| **Reward Generalizability ($n = 20$)** | | | | |
| AVRIL Batch | 8.923 | 1.197 | 1.152 | -180.305 |
| AVRIL Batch XL | 8.809 | **1.099** | **0.994** | -180.770 |
| AVRIL SRL | 8.732 | 1.124 | 1.030 | -181.487 |
| AVRIL Single | 9.017 | 1.418 | 1.280 | -178.544 |
| MSRD-Offline | 8.368 | 1.403 | 1.274 | -179.694 |
| DROID (ours) | **8.048** | 1.441 | 1.336 | **-175.528*** |

**Lunar Lander**

| Benchmark Method | Frechet Distance | Undirected Hausdorff | Log Likelihood | GT Reward |
|---|---|---|---|---|
| **Diverse Demonstration Modeling ($n = 40$)** | | | | |
| BC Batch | 5.496 | 5.400 | -122.774 | -24.613 |
| BC Single | 3.847 | 3.787 | -124.510 | -19.736 |
| Diffusion | 4.213 | 4.174 | -299.733 | -50.409 |
| BeT | 3.980 | 3.926 | -277.258 | -54.614 |
| AVRIL Batch | 6.209 | 5.436 | -275.345 | -72.539 |
| AVRIL Batch XL | 1.380 | 1.342 | -154.998 | -7.132 |
| AVRIL SRL | 2.012 | 1.919 | -57.810 | -13.275 |
| AVRIL Single | 1.488 | 1.459 | -151.956 | **-1.329** |
| MSRD-Offline | 3.643 | 3.353 | -247.261 | -30.652 |
| DROID | **1.153** | **1.063*** | **-54.941** | -6.637 |
| **Policy Generalizability ($n = 20$)** | | | | |
| BC Batch | 5.186 | 5.020 | -198.297 | -21.872 |
| BC Single | 3.434 | 3.356 | -197.667 | -17.448 |
| Diffusion | 4.165 | 4.146 | -299.734 | -47.071 |
| BeT | 3.755 | 3.717 | -277.258 | -47.544 |
| AVRIL Batch | 5.621 | 5.131 | -275.345 | -64.985 |
| AVRIL Batch XL | 1.400 | 1.374 | -185.511 | -4.492 |
| AVRIL SRL | 2.080 | 1.979 | -873.198 | -12.429 |
| AVRIL Single | 1.544 | 1.470 | -182.000 | -5.313 |
| MSRD-Offline | 3.476 | 3.244 | -251.386 | -28.495 |
| DROID (ours) | **1.158*** | **1.061*** | **-139.133** | **-2.020** |
| **Reward Generalizability ($n = 20$)** | | | | |
| AVRIL Batch | 4.042 | 3.835 | **-326.569** | -28.436 |
| AVRIL Batch XL | 4.079 | 4.049 | -378.684 | -66.989 |
| AVRIL SRL | 3.474 | 3.332 | -327.652 | -27.960 |
| AVRIL Single | 4.305 | **3.159** | -328.189 | -26.141 |
| MSRD-Offline | 4.870 | 4.662 | -327.508 | -33.656 |
| DROID (ours) | **3.438** | 3.326 | -328.853 | **-25.271** |

desired goal point), Undirected Hausdorff Distance (how closely the generated path and expert path align), and Average Log Likelihood. We train each technique on a dataset of 117 distinct training sols (each with three waypoints: start point, midpoint, and goal point) from 37 RPs and hold out one sol per RP (37 sols) as a test dataset to evaluate generalization performance. Note data is extremely limited as each RP is associated with only 2-5 Sols, and we treat each RP as a unique expert strategy. We provide more experiment details in the supplementary.

**Q1: Diverse Demonstration Modeling.** We show in Table 2 that DROID is more successful at the imitation objective with respect to baseline approaches. DROID outperforms ($p < .05$) baselines by 95% on reaching the goal point (Final Distance) along with 20% better modeling on the strategic preference (Log Likelihood). Policy distillation ensures DROID accomplishes the task goal while allowing it to model expert waypoint preferences well.

**Q2: Policy Generalizability.** On a holdout set of unseen Sols, Table 2 shows that DROID achieves better ($p < .05$) performance on the Undirected Hausdorff (21%) along with Final distance (90%) compared to best baselines. Despite limited and heterogeneous data, DROID's learned policy generalizes well to achieve high performance and model expert preferences closely compared to baselines.

**Q3: Reward Generalizability.** Similar to the OpenAI Gym experiments, we evaluate how successful the learned reward functions are at inducing high-performing policies by training CQL policies. The results in Table 2 section "Reward Generalizability" show DROID's learned reward successfully induces policies with significantly better ($p < .05$) performance along the Undirected Hausdorff metric (12%) and the Final Distance metric (36%).

### 6.2.1 Qualitative Analysis

We visualize the task reward and strategy reward learned by DROID for a randomly selected Sol in supplementary Figure 6 to understand RPs' preferences. We observe that the task reward encodes the common goal of converging at the goal point by giving high rewards to the goal area. In contrast, the strategy reward correctly identifies the midpoint preference. The illustration shows DROID can successfully decompose the shared goal along with modeling latent strategies on unseen domains.

Table 2: This table shows performance comparisons of DROID and baselines in MPP. Bold denotes the best-performing model. * denotes $p < .05$ significance against the second-best method.

| Benchmark Methods | Undirected Hausdorff | Distance from Waypoint | Final Distance | Log Likelihood |
|---|---|---|---|---|
| **Diverse Demonstration Modeling ($n = 114$)** | | | | |
| BC Batch | 8.924 | 9.408 | 5.020 | -43.192 |
| BC Single | 14.824 | 9.245 | 13.200 | -192.530 |
| Diffusion | 11.397 | 8.421 | 9.591 | -43.894 |
| AVRIL Batch | 11.825 | 10.517 | 1.428 | -9.129 |
| AVRIL Batch XL | 10.066 | 8.939 | 1.643 | -65.306 |
| AVRIL SRL | 9.096 | 8.401 | 6.313 | -26.425 |
| AVRIL Single | 10.099 | 8.445 | 7.356 | -15.662 |
| MSRD-Offline | 8.162 | 7.580 | 4.476 | -27.667 |
| DROID (ours) | **6.780**\* | **4.592** | **0.070**\* | **-7.261**\* |
| **Policy Generalizability ($n = 49$)** | | | | |
| BC Batch | 7.791 | 9.562 | 3.197 | -39.923 |
| BC Single | 14.570 | 12.591 | 2.755 | -186.310 |
| Diffusion | 10.886 | 8.775 | 3.506 | -43.175 |
| AVRIL Batch | 8.387 | 10.020 | 3.878 | -22.623 |
| AVRIL Batch XL | 11.032 | 8.589 | 4.205 | -61.561 |
| AVRIL SRL | 8.885 | **6.082** | 3.935 | -23.756 |
| AVRIL Single | 8.336 | 8.888 | 5.951 | **-17.357** |
| MSRD-Offline | 9.732 | 8.886 | 7.462 | -148.964 |
| DROID (ours) | **6.144**\* | 6.407 | **0.277**\* | -18.483 |
| **Reward Generalizability ($n = 49$)** | | | | |
| AVRIL Batch | 9.385 | 9.502 | 1.823 | -15.501 |
| AVRIL Batch XL | 9.396 | 9.502 | 1.925 | -15.653 |
| AVRIL SRL | 10.341 | 9.753 | 0.928 | -16.475 |
| AVRIL Single | 10.456 | 10.407 | 0.676 | -18.039 |
| MSRD-Offline | 8.799 | 8.867 | 1.268 | **-14.515** |
| DROID (ours) | **8.240**\* | **8.764** | **0.433**\* | -15.050 |

## 6.3 Ablation

In this section, we perform ablation studies to evaluate the utility of different components of DROID. Ablation 1-6 corresponds to the following: 1) DROID without AVRIL improvements 1 and 2; 2) DROID without AVRIL Improvement 1; 3) DROID without AVRIL Improvement 2; 4) DROID without distillation; 5) DROID without policy distallation; 6) DROID without reward distillation.

Supplementary Table 7 shows the results of our ablation study in Cartpole and MPP domains. DROID outperforms Ablation 4 on all metrics, showing the importance of distillation across different strategies on the Diverse Demonstration Modeling task. Ablation 5 overfits to the demonstrations and achieves poorer generalization performance compared to DROID, which demonstrates the benefit of policy distillation to improve policy generalizability. Ablation 6 achieves worse Log Likelihood, suggesting that reward distillation's ability to share knowledge among demonstrations helps DROID understand expert behaviors. Ablation 1-3 perform worse than DROID in all tasks except Log likelihood in MPP, demonstrating that our two AVRIL enhancements are effective in improving DROID's ability to learn demonstrator's latent intention. Overall, our ablation study highlights the importance of both reward and policy distillation, as well as the two enhancements, in achieving state-of-the-art performance. More ablation study details are included in the supplementary.

## 7 Conclusion, Limitations, and Future Work

In this paper, we introduce an OLfD technique, DROID, that expands the applicability of OLfD with heterogeneous and limited data by a novel decomposition of the policy and reward models. Our results on both simulated and real-world data demonstrate that DROID outperforms SOTA methods, particularly in capturing difficult-to-articulate knowledge from rover path planners at NASA.

There are several limitations with DROID. Firstly, DROID assumes experts have stationary preferences across demonstrations. Secondly, DROID assumes access to the number of strategies and the strategy label for each demonstration, which may be non-trivial to obtain. Thirdly, in the MPP domain, we extract nine features, which may not encompass all features an RP considers. In future work, we plan to explore modeling nonstationary strategies for demonstrators, test DROID's utility with RP's workflow, explore automatic feature extraction for Mars terrain, and relax the assumption about known strategy labels with behavior clustering [30] or online inference [29].

**Acknowledgments**

We wish to thank our reviewers for their valuable feedback in revising our manuscript. This work was supported by the National Institutes of Health (NIH) under Grant 1R01HL157457, by a NASA Early Career Fellowship under Grant 80NSSC20K0069, and by the National Science Foundation (NSF) under Grant #2219755.

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

## A Offline LfD Enhancements Detail

AVRIL considers a distribution over the reward function and approximates the posterior, $p(R|\mathcal{D})$, with a variational distribution, $q_\phi(R)$. It is trained by maximizing the Evidence Lower BOund (ELBO), shown in Equation 8, where $p(R)$ is the prior distribution for the reward function and $\pi_E$ is the expert policy. The second equation follows by the assumption of Boltzmann rationality of the demonstrator [67].

$$
\begin{aligned}
\text{ELBO}(\phi) &= \mathbb{E}_{q_\phi}\left[\log p(\mathcal{D}|R)\right] - D_{\text{KL}}(q_\phi(R)||p(R)]) \\
&= \mathbb{E}_{q_\phi}\left[\sum_{(s,a)\in D} \log \frac{\exp\left(\beta Q_R^{\pi_E}(s,a)\right)}{\sum_{b\in A}\exp\left(\beta Q_R^{\pi_E}(s,b)\right)}\right] - D_{\text{KL}}(q_\phi(R)||p(R)])
\end{aligned}
\tag{8}
$$

Directly optimizing ELBO is not feasible as the gradient of $Q_R^{\pi_E}(s,a)$ with respect to $\phi$ is intractable. Therefore, AVRIL introduces a second variational approximation for $Q_R^{\pi_E}(s,a)$ with $Q_\theta(s,a)$ and ensures the variational reward distribution, $q_\phi(R)$, is consistent with the variational Q function, $Q_\theta(s,a)$, by Bellman equation, i.e., $R(s,a) = \mathbb{E}_{s',a'\sim\pi}[Q_R^\pi(s,a) - \gamma Q_R^\pi(s',a')]$.

Here, we present a lemma to show how AVRIL Enhancement 1 (i.e., extending KL-divergence regularization on all actions) impacts reward learning. This enhancement could be viewed as a data augmentation technique encouraging a small distance to the prior distribution for any action. We formalize the intuition in Lemma 1.

**Lemma 1.** *Assume the prior reward distribution, $p(R(s,a))$, is a Gaussian distribution partitioned on each state and action pair, minimizing $L_{KL}$ results in $q_\phi(R(s,a)) = p(R(s,a))$ for each operated $(s,a)$.*

Following Lemma 1 and our extended operation over $b \in \mathbb{A}$, we have the following observation.

**Corollary 1.1.** *Assume we choose the prior reward distribution $p(R(s,a))$ to be Standard Gaussian distribution, $\mathcal{N}(0,1)$. for $s \in \mathcal{D}, b \in \mathbb{A}$ s.t. $(s,b) \notin \mathcal{D}$, optimizing $L_{AVRIL}$ leads to $\mu_\phi(s,b) = 0$ and $\sigma_\phi^2(s,b) = 1$. The proof follows immediately by observing $q_\phi(R(s,b))$ only gets gradient from $L_{KL}$ and the optimal solution of $L_{KL}$ is that $\mu_\phi(s,a) = 0$ and $\sigma_\phi^2(s,a) = 1$.*

## B MPP Heterogeneity Analysis Details

In our analysis, we seek to compare the variance of path features within each RP to the variance of path features across RPs, as this would help quantify how diverse expert demonstrations are. The demonstration for each RP is multivariate and not normal (we tested for normality and homoscedasticity), necessitating the use of the PERMANOVA test, which is non-parametric and can compare multivariate data. More specifically, it tests the null hypothesis that the centroid and dispersion for the two groups are equivalent. To apply this test to the RP data, we tested each possible pair of RPs to see which ones have statistically significant differences in their distributions. The Bonferroni-Holm method was used to account for the fact that many hypothesis tests are performed.

## C Experimental Details

For fair comparisons with all baseline techniques, we share the same network architecture for each policy and reward with two hidden layers of 64 units, along with GELU activation functions. The training is with Adam Optimizer for 1000 iterations. For downstream policies, we train offline Conservative Q Learning [66] for 1000 iterations. Conservative Q-learning is an offline RL algorithm that guards against overestimation while avoiding explicitly constructing a separate behavior model. We leverage several improvements, including Dueling Double Q Networks and Distributional RL from Rainbow [68] to improve the CQL training. We list hyperparameters used in all algorithms in Table 3.

| Hyperparameters | Values |
|---|---|
| Training Iterations | 1000 |
| Learning Rate | 0.0001 |
| State Only Reward | False |
| State Dim | 4, 9 |
| Action Dim | 2 |
| Gamma | 0.99 |
| Lambda | 1.0 |
| Train Test Split | 0.8 |
| Min Number of Test Sols | 1 |
| Linear Reward | False |
| Offline CQL Training Itrs | 1000 |
| Strategy Reward Regularization Coeffficient (MSRD and DROID) | 0.01 |
| Strategy Q Function Regularization Coeffficient (DROID) | 0.001 |

Table 3: This table shows the hyperparameters we use for DROID and all benchmark algorithms. All values separated with commas are for CartPole (LL) and MPP, respectively.

To showcase the statistical significance of our results on the Cartpole, LL, and MPP domains, we perform tests for normality and homoscedasticity and find that our metrics do not satisfy the assumptions of the parametric ANOVA test. Thus, we instead perform a non-parametric Friedman test followed by a posthoc Nemenyi–Damico–Wolfe (Nemenyi) test. We show significance by aligning demonstration strategies between treatments (benchmark techniques).

## C.1 CartPole / Lunar Lander

### C.1.1 Video Demonstrations

We include demonstrations of heterogeneous behaviors along with each technique's learned policies in CartPole and Lunar Lander in the link: https://tinyurl.com/droidcartpolevideos.

### C.1.2 Metrics

Here, we describe the motivation behind each of the metrics, evaluated from rollouts of the policies with respect to expert demonstrations.

1. Frechet Distance [62]: Compare the spatial and temporal differences of the trajectory from the agent's policy with the expert trajectory to quantify how well the agent captures the motion pattern of the expert.

2. KL Divergence [63] (CartPole): By estimating the state distribution within a trajectory by the kernel density estimator [69], KL divergence quantifies how well the learned policies state visitation matches the expert's.

3. Undirected Hausdorff Distance [64]: This measures the maxima between the two Directed Hausdorff distances: one mapping our learned policy's trajectory to the expert trajectory, and the other mapping the expert trajectory to our learn policy's trajectory. This metric studies how far the agent's trajectory is from the expert's trajectory.

4. Average Log-Likelihood: This measures the likelihood of expert demonstration under the learned policy.

5. Ground Truth Rewards (Lunar Lander): The environment reward for Lunar Lander (all benchmarks in CartPole achieve near-maximal reward).

### C.1.3 Analysis

We show Friedman Chi-square and Post-hoc Nemenyi statistical test metrics in CartPole for the three experiments (Diverse Demonstration Modeling, Policy Generalizability, and Reward General-

Table 4: This table shows the APA-style statistical test results for Friedman ($\alpha = 0.05$, d.o.f.=3) and Posthoc Nemenyi ($\alpha = 0.05$) of DROID with respect to baselines in Cartpole. All reported test statistics are significant other than the italicized metrics (if the Friedman test results are insignificant, no posthoc analysis is performed).

**CartPole**

| Benchmark Method | KL Divergence | Frechet Distance | Undirected Hausdorff | Log Likelihood |
|---|---|---|---|---|
| **Diverse Demonstration Modeling ($n = 40$)** | | | | |
| Friedman | 174.82 | 222.92 | 240.52 | 339.68 |
| DROID vs BC Batch | 5.20 | 4.80 | 6.06 | *8.71* |
| DROID vs BC Batch Large | 5.13 | 4.73 | 6.13 | *9.23* |
| DROID vs Diffusion | 4.62 | 11.04 | 11.82 | 3.14 |
| DROID vs BeT | 9.42 | 10.71 | 11.52 | 1.66 |
| DROID vs AVRIL Batch | *2.84* | 3.36 | 4.21 | 2.66 |
| DROID vs AVRIL Batch XL | 3.32 | 3.58 | 4.28 | 4.06 |
| DROID vs AVRIL SRL | 6.87 | 5.83 | 6.06 | *7.87* |
| DROID vs AVRIL Single | 5.13 | 4.99 | 5.24 | *1.85* |
| DROID vs MSRD-Offline | *1.77* | 3.05 | 3.29 | 5.50 |
| **Policy Transferability ($n = 20$)** | | | | |
| Friedman | 82.53 | 115.51 | 122.72 | 148.93 |
| DROID vs BC Batch | 3.97 | 4.23 | 3.92 | *5.12* |
| DROID vs BC Batch Large | 2.77 | 3.86 | 4.28 | *5.69* |
| DROID vs Diffusion | 6.37 | 8.20 | 8.25 | 2.77 |
| DROID vs BeT | 6.74 | 7.99 | 8.04 | 1.62 |
| DROID vs AVRIL Batch | *2.30* | 3.39 | 3.86 | *1.36* |
| DROID vs AVRIL Batch XL | *2.25* | 3.45 | 3.34 | 2.56 |
| DROID vs AVRIL SRL | 4.96 | 4.70 | 4.07 | *4.49* |
| DROID vs AVRIL Single | *2.45* | 2.66 | 2.98 | 0.78 |
| DROID vs MSRD-Offline | *1.62* | 2.72 | 2.94 | 3.71 |
| **Reward ($n = 20$)** | | | | |
| Friedman | *2.22* | *0.54* | *0.3* | 13.38 |
| DROID vs BC Batch | N/A | N/A | N/A | N/A |
| DROID vs BC Batch Large | N/A | N/A | N/A | N/A |
| DROID vs Diffusion | N/A | N/A | N/A | N/A |
| DROID vs BeT | N/A | N/A | N/A | N/A |
| DROID vs AVRIL Batch | N/A | N/A | N/A | 2.82 |
| DROID vs AVRIL Batch XL | N/A | N/A | N/A | 2.87 |
| DROID vs AVRIL SRL | N/A | N/A | N/A | 3.01 |
| DROID vs AVRIL Single | N/A | N/A | N/A | 2.08 |
| DROID vs MSRD-Offline | N/A | N/A | N/A | 3.43 |

izability, c.f. main paper Results Section Q1-Q3) in Table 4. In the training task, DROID generates rollouts that align closer with expert behaviors, evidenced by stronger Undirected Hausdorff performance. Likewise, DROID does significantly better on Frechet and Undirected Hausdorff distance compared to the best baselines in both the Demonstration modeling and Policy Transferability tasks. Common reward-policy distillation helps guide DROID's policies and rewards to better model expert preferences and, thus, better capture diversity in expert behaviors.

Likewise, we show the statistical test results for Lunar Lander in Table 5. DROID demonstrates clear superiority over several baseline methods. For instance, when compared to BC Batch and BC Batch Large, DROID achieves significantly better Frechet Distance values. This indicates that DROID's trajectory predictions align more closely with ground truth demonstrations than those of the standard BC baseline methods. Similarly, in the "Policy Transferability" benchmark, DROID consistently outperforms its counterparts. Compared to BC Batch, DROID attains substantially lower Frechet Distance and Undirected Hausdorff Distance than the best baselines in AVRIL SRL and AVRIL Batch XL. This suggests that DROID's explicit strategy decomposition and knowledge sharing improve overall performance.

Table 5: This table shows the APA-style statistical test results for Friedman ($\alpha = 0.05$, d.o.f.=3) and Posthoc Nemenyi ($\alpha = 0.05$) of DROID with respect to baselines in Lunar Lander. All reported test statistics are significant other than the italicized metrics (if the Friedman test results are insignificant, no posthoc analysis is performed).

**Lunar Lander**

| Benchmark Method | Frechet Distance | Undirected Hausdorff | Log Likelihood | Ground Truth |
|---|---|---|---|---|
| **Diverse Demonstration Modeling ($n = 40$)** | | | | |
| Friedman | 171.61 | 170.31 | 322.40 | 207.27 |
| DROID vs BC Batch | 6.94 | 7.98 | 3.84 | 4.91 |
| DROID vs BC Batch Large | 6.24 | 6.13 | 4.10 | 3.91 |
| DROID vs Diffusion | 8.97 | 9.16 | 12.56 | *8.23* |
| DROID vs BeT | 8.9 | 9.08 | 11.08 | *8.75* |
| DROID vs AVRIL Batch | *3.40* | 4.14 | 5.28 | *0.37* |
| DROID vs AVRIL Batch XL | *1.92* | 2.88 | 6.68 | *0.18* |
| DROID vs AVRIL SRL | 4.73 | 5.39 | 0.15 | *3.32* |
| DROID vs AVRIL Single | *3.43* | *2.73* | *5.80* | 0.44 |
| DROID vs MSRD-Offline | 8.27 | 7.90 | 9.60 | 6.06 |
| **Policy Transferability ($n = 20$)** | | | | |
| Friedman | 71.93 | 76.90 | 122.76 | 87.25 |
| DROID vs BC Batch | 4.49 | 2.66 | *1.51* | *3.03* |
| DROID vs BC Batch Large | 3.86 | 3.92 | *1.62* | *3.08* |
| DROID vs Diffusion | 6.27 | 6.74 | 6.68 | 5.69 |
| DROID vs BeT | 5.74 | 5.48 | 5.43 | 6.16 |
| DROID vs AVRIL Batch | 2.14 | 2.04 | *0.89* | *0.78* |
| DROID vs AVRIL Batch XL | 2.67 | 1.98 | *1.36* | *0.11* |
| DROID vs AVRIL SRL | 3.08 | 2.66 | 7.15 | *2.87* |
| DROID vs AVRIL Single | 2.40 | 2.30 | *1.15* | *1.62* |
| DROID vs MSRD-Offline | 5.33 | 5.64 | 3.97 | 4.44 |
| **Reward ($n = 20$)** | | | | |
| Friedman | *25.46* | *32.02* | *29.15* | *60.94* |
| DROID vs BC Batch | N/A | N/A | N/A | N/A |
| DROID vs BC Batch Large | N/A | N/A | N/A | N/A |
| DROID vs Diffusion | N/A | N/A | N/A | N/A |
| DROID vs BeT | N/A | N/A | N/A | N/A |
| DROID vs AVRIL Batch | N/A | N/A | N/A | N/A |
| DROID vs AVRIL Batch XL | N/A | N/A | N/A | N/A |
| DROID vs AVRIL SRL | N/A | N/A | N/A | N/A |
| DROID vs AVRIL Single | N/A | N/A | N/A | N/A |
| DROID vs MSRD-Offline | N/A | N/A | N/A | N/A |

## C.2 Mars Path Planning

### C.2.1 Domain Introduction

Exploring Mars has been a fascinating and challenging endeavor for space agencies worldwide. The Curiosity Rover is the longest active autonomous vehicle NASA has sent to Mars to study the climate, geology, and potential habitability of the planet [10]. The Rover has been in operation for the past ten years, and its path planning has been done by manual labor of Rover Planners (RPs) on Earth.

There are several factors RPs consider when designing paths, including the change in elevation, distance to the desired destination, uncertainty about missing data on the terrain, etc. We study a dataset of Curiosity Rovers curated paths from 154 sols (a sol being a Martian day, approximately 24.6 hours). We demonstrate in Section 4 of the main paper that there is significant heterogeneity between RP's paths. Each RP has a specific priority among safety, efficiency, risk, and mission constraints that inform their path design. This motivates us to design an autonomous path-planning approach that learns from these heterogeneous experts.

### C.2.2 Dataset Curation

The data consists of features that were created through a series of interviews with RPs, scientists, and engineers that ideally capture the decision-making process for rover path planning. The features were engineered to codify the reasoning behind the RP's decisions. For example, RPs would visually

analyze the terrain and map waypoints to avoid "rough" terrains without quantifiable measures of what is considered rough. We identify the following features to encode the mental models and strategies of RPs with considerations of risks, efficiency, safety, and mission requirements.

**Distance Feature –** The distance feature measures the percent added distance the rover must take by adding intermediate waypoints in relation to the direct distance between the start and end waypoints. The aim of this feature is to drive the rover's necessary additional distance. It is assumed that the path between waypoints is driven straight as RPs rarely drive curved paths and rather set additional intermediate points if the rover needs to avoid hazards between waypoints.

**Unknown Data Feature –** When constructing the height maps, data could be missing where the cameras cannot see terrain beyond a hill or obstructions like a large rock or the rover itself. By traversing terrains with missing data, the RP places the rover at a higher risk of damage. The design of the Unknown Data Feature is to minimize the distance the rover traverses over terrain without data. We compute the unknown data feature as the percent data missing in the height map for proposed trajectories.

**Roughness Feature –** Rover Planners ideally drive on relatively smooth surfaces, avoiding rough terrain that could potentially damage the rover's hardware. Similarly, Rover Planners also look to avoid terrain that is too soft to prevent a similar fate as Spirit getting stuck in sand [70]. Here, the roughness is computed as the difference of consecutive surface angles as the rover traverses to the goal point. The maximum roughness and the average roughness over proposed trajectories are measured to avoid large holes or rocks and minimize traveling on rough terrains.

**Pitch and Roll Feature –** Pitch and roll of the rover add another level of safety checks that ensure that the rover will not face terrains that risk the rover rolling over.

**Turning Trajectory –** We include the turning trajectory as a feature to track. This feature calculates the angle the rover must turn at intermediate waypoints. With this feature, the cost of taking sharp turns considers the rover's hardware and long-term health.

**Waypoint Grid Construction –** The 64x64-sized waypoint grid is constructed by scaling the terrain height map along each axis and sampling the terrain map height at each (x,y) coordinate in the scaled grid. We do so according to an inverse weighted distance from each point along the four nearest points with height map data. We perform this scaling to limit the size of the action space of possible waypoints.

$$H(x,y) = \frac{\frac{h_1}{d_1} + \frac{h_2}{d_2} + \frac{h_3}{d_3} + \frac{h_4}{d_4}}{\frac{1}{d_1} + \frac{1}{d_2} + \frac{1}{d_3} + \frac{1}{d_4}}$$

$H$ represents the height evaluated at each point in the gaming area. $(x,y)$ represent the corresponding coordinates, and $h_i, d_i$ represent the height and distance away from the evaluated point in the dataset, respectively.

### C.2.3 Description of Policy

The action space exists on the 64 by 64 discrete grid of 4096 possible successor waypoints, which was chosen to be large enough to have high precision when selecting waypoints. The average distance between grid points ranges from 0.01m to 0.7m for different Sols. We define our learned policy in Equation 9 from our learned Q-function $Q_\theta$.

$$\pi_\theta(s) = \max_{a \in A} Q_\theta(s, a) \tag{9}$$

As mentioned in the main paper, we consider the three-waypoint planning problem and therefore, an action, $a$, (i.e., the intermediate waypoint) determines the trajectory as from the current point to the intermediate waypoint and then from the intermediate waypoint to the ending waypoint. We calculate the features of the action (i.e., next waypoint) for each of the two segments of the trajectory (current point to next waypoint, and next waypoint to goal point).

Table 6: This table shows the APA-style statistical test results for Friedman ($\alpha = 0.05$, d.o.f.=3) and Posthoc Nemenyi ($\alpha = 0.05$) of DROID with respect to baselines in MPP. All reported test statistics are significant other than the italicized metrics (if the Friedman test results are insignificant, no posthoc analysis is performed).

**Mars Path Planning**

| Benchmark Methods | Undirected Hausdorff | Distance from Waypoint | Final Distance | Log Likelihood |
|---|---|---|---|---|
| **Diverse Demonstration Modeling ($n = 114$)** | | | | |
| Friedman | 112.98 | 44.91 | 408.32 | 786.47 |
| DROID vs BC Batch | 2.19 | *0.15* | 3.93 | 4.33 |
| DROID vs BC Batch Large | 2.29 | 0.67 | 14.45 | 4.23 |
| DROID vs Diffusion | 4.22 | *1.11* | 4.77 | 4.04 |
| DROID vs AVRIL Batch | 6.29 | 4.00 | 4.84 | 12.07 |
| DROID vs AVRIL Batch XL | 4.09 | *0.07* | 5.07 | 3.17 |
| DROID vs AVRIL SRL | 5.18 | *0.83* | 6.43 | 7.57 |
| DROID vs AVRIL Single | 6.19 | 3.86 | 12.74 | 12.16 |
| DROID vs MSRD-Offline | 2.30 | *0.24* | 10.08 | 6.51 |
| **Policy Transferability ($n = 49$)** | | | | |
| Friedman | 74.09 | *15.08* | 153.77 | 345.76 |
| DROID vs BC Batch | 2.14 | N/A | *3.83* | *1.49* |
| DROID vs BC Batch Large | 2.89 | N/A | *3.41* | *1.42* |
| DROID vs Diffusion | 2.55 | N/A | *3.73* | 3.91 |
| DROID vs AVRIL Batch | 4.15 | N/A | *7.43* | 8.89 |
| DROID vs AVRIL Batch XL | 3.11 | N/A | *2.81* | 3.28 |
| DROID vs AVRIL SRL | 3.73 | N/A | *2.99* | 6.27 |
| DROID vs AVRIL Single | 4.22 | N/A | *7.14* | 8.63 |
| DROID vs MSRD-Offline | 3.55 | N/A | *2.72* | 3.84 |
| **Reward ($n = 49$)** | | | | |
| Friedman | 65.42 | 172.63 | 173.50 | 129.79 |
| DROID vs BC Batch | N/A | N/A | N/A | N/A |
| DROID vs BC Batch Large | N/A | N/A | N/A | N/A |
| DROID vs Diffusion | N/A | N/A | N/A | N/A |
| DROID vs AVRIL Batch | 2.73 | 6.16 | 6.51 | *0.21* |
| DROID vs AVRIL Batch XL | 6.10 | 7.96 | 8.10 | *0.34* |
| DROID vs AVRIL SRL | 5.97 | 7.69 | 7.94 | 2.88 |
| DROID vs AVRIL Single | 3.16 | 6.05 | 6.56 | 4.59 |
| DROID vs MSRD-Offline | 2.18 | *0.32* | 3.59 | *2.48* |

### C.2.4 Metrics

Here, we include further description of the metrics we study in the MPP problem:

1. Average Distance from Midpoint: The average distances from our policies' predicted waypoints to the demonstrated waypoints.

2. Distance from Endpoint: The average distance from the final waypoint selected by the path generated by each technique to the goal point.

3. Undirected Hausdorff Distance [64]: This metric measures the maxima between the Directed Hausdorff distances mapping both our learned policy's set of waypoints to the expert waypoints and vice-versa.

4. Average Log Likelihood: This metric measures the likelihood of expert demonstration under the learned policy.

### C.2.5 Analysis

We showcase the specific metrics that DROID outperforms baseline techniques on for MPP in Table 6. Rather than assuming homogeneity across demonstrations or discarding data to design a personalized policy for each RP, DROID takes advantage of per-RP modeling and knowledge sharing to significantly outperform three out of four metrics in the Diverse Demonstration modeling benchmark. On the policy generalization benchmark, DROID can also model the latent objectives from diverse experts to design a trajectory in unseen Sols that align closer to the expert's true path while successfully capturing the high-level common task goal. Lastly, DROID is the only technique

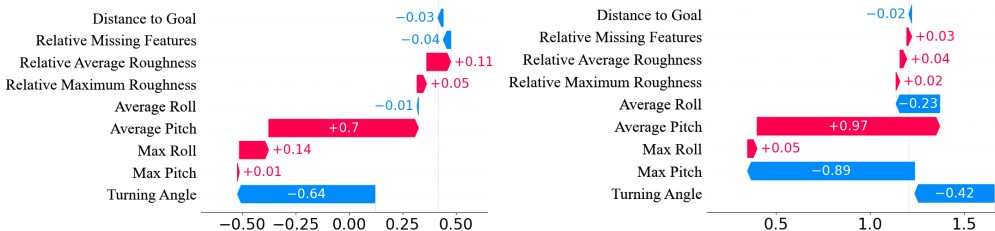

Figure 4: This figure shows Shapley values (the contribution of each of the composite features to the model's reward estimate) for the learned rewards of RP 1 (left) and 5 (right) evaluated on Sol 2030's demonstrated path.

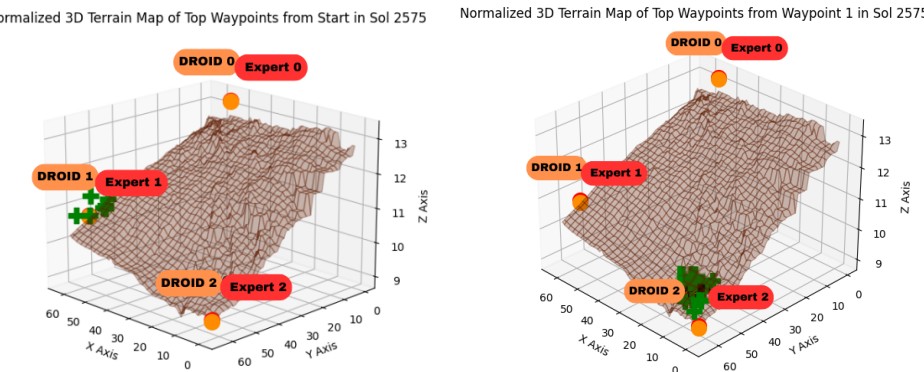

Figure 5: These figures show DROID's policy outputs on a terrain map to plan for the next waypoint from the Start point (left) and Waypoint 1 (right). The orange spheres represent the selected waypoints of DROID and the expert. Highlighted in green above the terrain map are the top 10 highest-rated successor waypoints. The orange labels correspond to DROID's found waypoints, and the red labels correspond to the expert demonstration's waypoints.

to show significantly better performance on downstream reward transfer, indicating the learned reward is a more useful encoding of an expert's latent objective and can be used to better interpret the salient features of a given expert.

### C.2.6 Additional Qualitative Analysis

In this section, we discuss the additional contributions of DROID to the goal of interpreting expert decision-making and how it is valuable in the domain of path planning for the Mars Curiosity Rover and future missions.

**Feature Contribution Analysis** We perform a Shapley value analysis [71] on each RP's learned reward function. The analysis measures how adding a feature would change the prediction output and is helpful in comparing the relative importance different RPs place on features, even on Sols they have not explicitly planned on. As shown in Figure 4, we evaluated two randomly selected RPs (1 and 5) on Sol 2030. We observe that all drivers value that the path has a low pitch. However, RP 1 prefers to have a smaller turning angle, while RP 5 values a lower pitch. This demonstrates how DROID can model heterogeneous strategies and quantify the influence of specific features on the modeled objective function. We can identify why certain RPs like or dislike a given path and understand which features contribute to that assessment.

**Learned Reward and Policy** First, we analyze our learned Q-function and study how it can provide insight into the decision-making process of our model. We showcase in Figure 5 how our technique can highlight the top 10 highest-rated successor waypoints from the start and midpoint

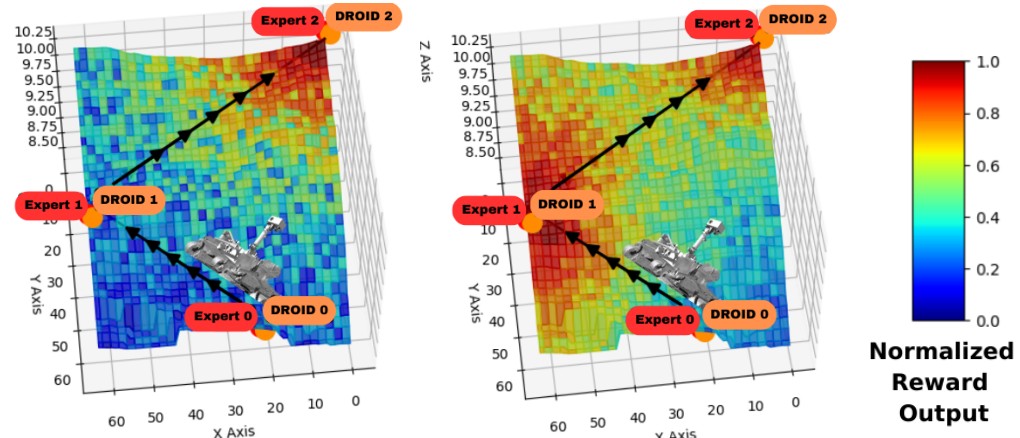

Figure 6: This figure visualizes actual 3D Mars terrain map with DROID's mean estimate of the learned task (left) and strategy (right) reward for Sol 2163. X-axis and Y-axis correspond to the surface coordinates, Z-axis corresponds to elevation, and heatmap coloring is the normalized reward output. The black line with arrows interlayed represents the path of the rover. The orange labels are DROID's found waypoints and the red labels are ground-truth demonstration waypoints. Point 0 is the starting point, Point 1 is the intermediate waypoint, and Point 2 is the final waypoint selected.

Table 7: This table shows the ablation performance of DROID in Cartpole (Left) and MPP (Right). Bold indicates the best-performing model of the metric.

**CartPole**

| Benchmark Method | KL Divergence | Frechet Distance | Undirected Hausdorff | Log Likelihood |
|---|---|---|---|---|
| **Diverse Demonstration Modeling ($n = 40$)** | | | | |
| Ablation 1 | 6.758 | 0.492 | 0.340 | -63.926 |
| Ablation 2 | 8.250 | 0.716 | 0.608 | -68.511 |
| Ablation 3 | 6.718 | 0.532 | 0.387 | -63.728 |
| Ablation 4 | 9.756 | 0.992 | 0.768 | -81.057 |
| Ablation 5 | **5.868** | **0.420** | 0.263 | -42.553 |
| Ablation 6 | 6.244 | 0.444 | 0.298 | -92.006 |
| DROID (ours) | 6.047 | 0.425 | **0.261** | **-37.948** |
| **Policy Generalizability ($n = 40$)** | | | | |
| Ablation 1 | 7.253 | 0.504 | 0.358 | -66.059 |
| Ablation 2 | 8.889 | 0.743 | 0.654 | -71.426 |
| Ablation 3 | 6.455 | 0.509 | 0.316 | -65.631 |
| Ablation 4 | 10.045 | 1.033 | 0.797 | -84.323 |
| Ablation 5 | **4.632** | 0.419 | 0.276 | -45.663 |
| Ablation 6 | 6.626 | 0.487 | 0.284 | -92.680 |
| DROID (ours) | 5.271 | **0.412** | **0.207** | **-38.057** |

**MPP**

| Benchmark Methods | Undirected Hausdorff | Distance from Waypoint | Final Distance | Log Likelihood |
|---|---|---|---|---|
| **Diverse Demonstration Modeling ($n = 40$)** | | | | |
| Ablation 1 | 4.871 | 1.557 | 8.391 | -10.157 |
| Ablation 2 | 6.084 | 0.288 | 7.575 | -10.104 |
| Ablation 3 | 7.126 | 0.571 | 7.287 | -8.431 |
| Ablation 4 | 6.720 | 0.209 | 7.441 | -8.419 |
| Ablation 5 | **3.910** | 4.014 | 7.498 | -225.212 |
| Ablation 6 | 5.556 | 6.783 | 9.389 | -14.479 |
| DROID | 4.592 | **0.070** | 6.780 | **-7.261** |
| **Policy Generalizability ($n = 40$)** | | | | |
| Ablation 1 | 8.086 | 1.842 | 8.945 | -15.010 |
| Ablation 2 | 8.071 | 1.615 | 8.331 | -16.334 |
| Ablation 3 | 9.318 | 3.933 | 9.644 | -16.503 |
| Ablation 4 | 8.518 | 0.576 | 7.744 | **-11.391** |
| Ablation 5 | 6.162 | 7.295 | 9.537 | -13.610 |
| Ablation 6 | 8.026 | 9.078 | 9.246 | -30.037 |
| DROID | **6.144** | **0.277** | **6.407** | -18.483 |

positions, respectively. Providing multiple options aligned with the expert's latent preference could be beneficial for a future assistive tool for RPs. Expert drivers at NASA can also study DROID's recommended waypoint and similar waypoints that are rated highly by our model. If an expert disagrees with the best action identified by DROID, we can find several additional options that align with that expert's latent preferences.

We examine the uncertainty in the reward predictions of our model by plotting the standard deviation of the strategy reward posterior. As shown in Figure 7, we can estimate how uncertain our model is about different parts of the terrain due to the limited coverage of the dataset. Intuitively, areas of the state space where the demonstrations have not covered, such as the edges of the terrain, have a higher estimate of uncertainty.

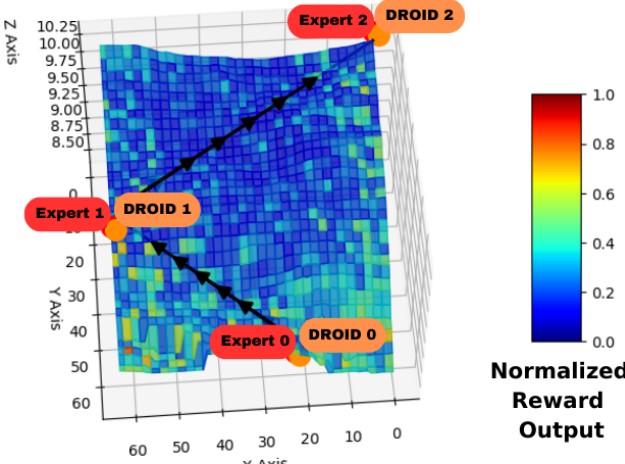

Figure 7: This figure shows a heatmap of Sol 2163 of DROID's Strategy Reward log standard deviation estimate, where higher value represents greater uncertainty in the reward estimation.

**Proposed Application to NASA**  The explainability provided by DROID has significant potential application as a supplementary planning tool for interplanetary exploration. By leveraging a Shapely value analysis of the importance of different factors, such as the change in elevation and uncertainty about missing data on the terrain, rover planners can gain a deeper understanding of the objective function our algorithm extracts for modeling rover path planning. Therefore, DROID can explain what features contribute most to its perceived estimate of any human or AI-designed path.

DROID may have an application to inform the design of future missions by providing more insight into the limitations of the rover and the types of environments in which it is best suited to operate. As shown on the Curiosity Rover dataset, DROID could be applied to give rapid feedback about paths that best avoid sharp rocks (Rough Terrain), which may damage the open holes on the rover's wheels [9], thus improving the longevity of the rover. Similarly, with additional data, such as orbital satellite imagery, our approach could be used to evaluate the value of landing sites [57] by studying our learned RP objective function on constructed terrain maps.

Furthermore, the ability to model different strategies taken by human drivers could potentially be used in the future by JPL in developing training programs. We hope a future application of DROID would be to capture difficult-to-articulate tribal knowledge among rover planners and identify the most important features to trainees. We can describe implicitly understood knowledge to help train new drivers at NASA faster and with greater efficiency. By letting DROID explain which features contribute most to the underlying latent RP strategy, human drivers can better understand what features to consider when navigating other extraterrestrial terrains.

Our hope is that DROID lessens the burden for operators to plan out daily schedules for rovers (since it performs automated path planning that better optimizes operator preferences). Moreover, the algorithm's ability to reason under uncertainty makes it particularly useful for fast path planning inference, even when occluded information exists from cameras or other sensors. With the DROID's ability to learn diverse expert strategies and plan under uncertainty/occlusions, our algorithm could further advance autonomous rover exploration.

# D   Additional Related Works

In this section, we describe additional related works regarding offline path planning under uncertainty and navigation beyond the MDP setting.

**Algorithm 1:** Dual Reward and policy Offline Inverse Distillation (DROID)

---

**Input** : Training iterations $E$, number of strategies $M = |D|$, demonstration dataset for all strategies $\mathcal{D} = \{\mathcal{D}_j\}_{j=1}^M$, learning rate $\alpha$.
**Output** : Learned policy set $\Pi$, rewards functions $\mathcal{R}$
**Initialize:** Initialize all reward function and policy parameters $\Theta = \{\theta_{\text{Task}}, \{\theta_{\text{S}-i}\}_{i=1}^M, \phi_{\text{Task}}, \{\phi_{\text{S}-i}\}_{i=1}^M\}\}$

1 **for** $i = 1$ *to* $E$ **do**
2    Zero Gradients for Shared Reward and Policy Parameters: $\theta_{\text{Task}}, \phi_{\text{Task}}$
3    **for** $j = 1$ *to* $M$ **do**
4      Sample a minibatch $(s, a) \sim \mathcal{D}_j$
5      Calculate $Q(s) = Q_{\text{task}}(s) + Q_{\text{S}-j}(s)$.
6      Combine loss terms to calculate overall loss $\mathcal{L}$ according to Equation 7
7        Calculate AVRIL loss $L_{\text{AVRIL}}(\theta, \phi)$ according to Equation 1 with the two improvements introduced in Equation 5 and 6
8        Calculate reward regularization $L_{\text{RD}}(\{\phi_{\text{S}-i}\}_{i=1}^M; \mathcal{D})$ according to Equation 3 and policy regularization terms $L_{\text{PD}}(\{\theta_{\text{S}-i}\}_{i=1}^M; \mathcal{D})$ according to Equation 4.
9      Calculate the gradient of $\mathcal{L}$ with respect to all parameters $\Theta$.
10      Update strategy-only reward parameters: $\phi_{\text{S}-j} \leftarrow \phi_{\text{S}-j} + \alpha \frac{\partial \mathcal{L}}{\partial \phi_{\text{S}-j}}$
11      Update strategy-only policy parameters: $\theta_{\text{S}-j} \leftarrow \theta_{\text{S}-j} + \alpha \frac{\partial \mathcal{L}L}{\partial \theta_{\text{S}-j}}$
12      Aggregate shared task reward gradients: $\Delta \phi_{\text{Task}} \leftarrow \Delta \phi_{\text{Task}} + \frac{\partial \mathcal{L}}{\partial \phi_{\text{Task}}}$
13      Aggregate shared task policy gradients: $\Delta \theta_{\text{Task}} \leftarrow \Delta \theta_{\text{Task}} + \frac{\partial \mathcal{L}}{\partial \theta_{\text{Task}}}$
14    Update shared reward and policy parameters with learning rate $\alpha$:
     $\theta_{\text{Task}} \leftarrow \theta_{\text{Task}} + \alpha \Delta \theta_{\text{Task}}, \phi_{\text{Task}} \leftarrow \phi_{\text{Task}} + \alpha \Delta \phi_{\text{Task}}$.

---

**Offline Learning**    Offline learning has been used to teach robots to perform tasks such as assembly, manipulation, and grasping in manufacturing settings [26]. It has also been used in healthcare to assist with tasks such as clinical diagnosis [26]. In medical diagnosis, offline LfD can be used to understand how expert clinicians diagnose a disease based on temporal indicators of key symptoms and medical reports. DROID's success in offline learning, particularly in complex domain settings, makes it useful for understanding an expert decision-making process without risking the safety of humans or expensive equipment.

**Path Planning Algorithm.**    Several works use human-inspired admissible heuristic functions to plan paths [72, 73]. Yet, these functions are handcrafted and require domain expertise to design. Model Predictive Path Integral (MPPI) is studied for local path following for rovers [74]. However, classical path planning approaches fail without a high-fidelity simulator [75, 76]. Other works look at the problem of path planning to maximize a reward function [77, 78, 79] under uncertainty. However, these techniques leverage exploration to obtain a better estimate of their cost function, which may not be feasible in offline learning. Our algorithm, DROID, learns heterogeneous preferences and policies directly from expert demonstrations without assuming a hand-designed reward function or a simulator.

**Generalization Performance of Navigation Algorithms.**    Another important factor in offline path planning is the generalization performance of the planning algorithm to novel terrains. To improve the generalization performance, existing work attempts to decouple the training of feature extraction and navigation blocks using Deep RL [80]. However, they perform online planning through 2D navigation to extract an attention map, which is not feasible in the offline setting. Additionally, Meng et al. [81] proposes a path planning algorithm that balances the trade-off between safety and efficiency under uncertainty. However, in contrast with DROID, these approaches do not generalize to unseen environments that contain new or additional obstacles.

**Global Path Planning for the Martian Domain.** Several prior works study path planning in the Martian domain but focus on local path planning (short-range egocentric navigation) and do not address global path planning: long-range end Hedrick et al. [56] proposes efficient Martian path planning and Rover-IRL [57] learns a local cost function from demonstration, but both fail to learn an RP objective function that can transfer to new and unexplored terrain spaces, a key challenge in the Mars domain [9]. Unlike previous path-planning approaches, DROID infers a global RP objective function that can transfer to planning downstream on unseen terrains.

# E Pseudocode for DROID

We present the pseudocode for DROID in Algorithm 1. For each training iteration $E$ (line 1), we iterate over all $M$ demonstrations (line 3). For each strategy, we sample a minibatch from the strategy's dataset, $\mathcal{D}_j$ (line 4). We then calculate the overall loss $\mathcal{L}$ (line 6), with which we calculate the gradient with respect to all parameters (line 9). We update the strategy-only reward and policy parameters (lines 10-11) and aggregate shared task reward and policy gradients (lines 12-13). After iterating over all $M$ strategies, we update the shared reward and policy parameters with the aggregated gradients (line 14).

