# OpenReview forum: "DROID: Learning from Offline Heterogeneous Demonstrations via Reward-Policy Distillation"
_robot-learning.org/CoRL/2023/Conference — CoRL 2023 Poster_

### Official Review · Reviewer_CWnj · 2023-07-19

**Confidence:** 4
**Originality:** Fair
**Technical Quality:** Good
**Clarity Of Presentation:** Very Good
**Impact:** 3

**Recommendation:**

Weak Accept: I recommend accepting the paper, but will not argue for my recommendation if the majority of other reviewers have a different opinion.

**Review:**

Strengths:
* Algorithm is presented in a very realistic setting with Mars path planning
* The motivation for the algorithm is clear, demonstrators have different styles
* Qualitative analysis clearly shows that the algorithm addresses the initial motivation
* Limitations are exhaustive enough


Weakness:
* Typically, adding explicit structure in learning algorithms can be addressed instead by adding modeling capacity which can implicitly model strategy/preferences for example. There is no mention of if increasing modeling capacity like number of model parameters also helps.
* The preferences are in relatively low-dimensional space in the examples presented. It would bolster their case if more complex datasets/tasks were included to show that this method more generally works in any LfD use case. It is unclear if the work is proposing their algorithm should be used in most LfD scenarios

**Quality Of The Limitations Section:**

Limitations are addressed clearly

**Questions For Rebuttal:**

* The Mars dataset was explicitly curated to research data heterogeneity and is thus already suited to DROID. Is DROID more generally suitable where it should always be used in LfD problems? Can you show this for more general datasets beyond Cartpole?
* It was unclear from the algorithm, do the number of strategies need to be set in the algorithm? If so that seems like a big drawback of the algorithm, how did you define this for the Cartpole problem?
* Two improvements are presented in Section 5.3, are there any ablations to show that these actually improve the method?

**Robotics Focus:**

Relevant but unlikely to deploy to hardware in near future

**Summary Of Paper:**

This work introduces a new offline learning from demonstrations algorithm, DROID, which includes an explicit modeling for different demonstrator strategy policy and rewards, a novel Gaussian paramerization is used to distill the rewards. Data heterogeneity and generalization are argued as the main motivation explicit model. Comparison to baselines are shown for modeling train and unseen demonstrations and also compared to expert behavior. A new Mars Curiosity Rover path planning dataset is introduced along with an analysis of how demonstrator strategies influence data heterogeneity. Results are also shown for a cartpole problem. An insightful qualitative analysis shows how DROID can model the different strategies in the Mars dataset.

**Summary Of Recommendation:**

Main rejection is how widely applicable this approach is, should it be applied to most LfD scenarios even when the presence of demonstrator strategies is not clear or explicit. Does this method scale with larger datasets and models where such preferences may be easily learned implicitly.

Given the additional experiments in the rebuttal phase,  I recommend a weak accept

---

### Official Review · Reviewer_CKuJ · 2023-07-19

**Confidence:** 3
**Originality:** Good
**Technical Quality:** Very Good
**Clarity Of Presentation:** Good
**Impact:** 4

**Recommendation:**

Weak Accept: I recommend accepting the paper, but will not argue for my recommendation if the majority of other reviewers have a different opinion.

**Review:**

Strengths
- The problem is well-motivated an appropriate approach is provided to address the problem setting
- The algorithm is evaluated on two domains, one of which is the Mars rover path-planning problem which is a great setting to ground the algorithm
- The performance of the algorithm outperforms the existing baselines meaningfully and with statistical significance.

Weaknesses:
- The paper considers only two additional baselines in this setting. This I believe isn't extensive enough and would encourage the authors to consider adding in additional points of comparison.

**Quality Of The Limitations Section:**

Limitations are addressed clearly

**Questions For Rebuttal:**

1. For the reward distillation (equation 3), is the assumption that each trajectory is labeled with the type of strategy used $c_t$ and the number of total strategies $M$ used known apriori?
2. There are two other points of comparison that can be considered to the approach as mentioned. Firstly, Imitation Learning approaches would not need a reward estimation and can learn from prior demonstrations directly. Methods such as BET or Diffusion based methods can deal well with multimodality. Additionally, offline RL methods could be learned in this setting with either goal-conditioned reward formulation or a heuristic reward (such as the last n timesteps of the demonstration data is successful). Could you compare how your method would fare against these methods?

**Robotics Focus:**

Highly relevant to robotics but no hardware experiments

**Summary Of Paper:**

This paper provides a method for offline learning from demonstrations. In particular, the authors choose to look at situations that are heterogeneous (i.e with diverse preferences and strategies) as well as look for generalization (adapting to unseen test settings). Their method is analyzed in two domains, on of which is the novel Mars rover path-planning problem. They show that their method outperforms baselines to a significant extent.

**Summary Of Recommendation:**

Overall, I reccomend the acceptance of this work. I think the work had algorithmic novelty and tested on some grounded, realistic problem settings.

---

### Official Review · Reviewer_tYgj · 2023-07-19

**Confidence:** 3
**Originality:** Very Good
**Technical Quality:** Very Good
**Clarity Of Presentation:** Very Good
**Impact:** 3

**Recommendation:**

Weak Accept: I recommend accepting the paper, but will not argue for my recommendation if the majority of other reviewers have a different opinion.

**Review:**

Overall I am positive on this paper and leaning towards acceptance. Experiments are conducted on a challenging and interesting real world dataset and while the reward and policy distillation approaches build on prior methods, they include interesting innovations such as using mean-field Gaussian distributions for reward distillation. I think the core strengths of this paper are in the challenging and novel experimental domain and the clear description of the methodological improvements over prior work in reward and policy distillation required to make DROID work in practice. One weakness of this work is the lack of appropriately controlled ablations in the experimental evaluation. For example, it would be nice to study DROID but with the exact same reward distillation approach as MSRD to understand the effect of the proposed changes in reward distillation.  Additionally, it would be helpful to study DROID but with the policy distillation process performed exactly as in AVRIL. Another very important comparison that should be included is to behavior cloning methods which do not directly infer reward functions, which are often very strong baselines for offline imitation learning. I appreciate the detailed experimental results otherwise though, and really enjoyed Figure 4.

**Quality Of The Limitations Section:**

Limitations are addressed clearly

**Questions For Rebuttal:**

1) Will the new Mars rover path planning dataset be made public? This would be a valuable contribution to the research community.
2) Would it be possible to add baselines which do not do direct reward inference but instead simply do behavior cloning (with both unimodal and multimodal policies)? These baselines often perform quite well in practice and would be helpful in establishing whether the complexities of DROID are truly necessary.
3) Please add ablation studies as mentioned above so the reader can better understand the impact of the design choices made in DROID.

**Robotics Focus:**

Sufficient demonstration on hardware

**Summary Of Paper:**

This paper proposes a new approach to handle heterogeneous demonstrations to learn generalizable policies in the context of offline imitation learning. The key contribution of the paper is achieving state of the art performance on a new and interesting dataset for Mars rover path planning and in proposing new and general reward and policy distillation methods for offline imitation learning.

**Summary Of Recommendation:**

Overall I enjoyed the paper and think the strong performance on the MARS path planning domain is quite interesting for the CoRL community. I am leaning towards acceptance, but would be further convinced by the ablations mentioned above and the addition of behavior cloning based baselines.

---

### Official Review · Reviewer_FG7m · 2023-07-24

**Confidence:** 3
**Originality:** Good
**Technical Quality:** Fair
**Clarity Of Presentation:** Fair
**Impact:** 3

**Recommendation:**

Weak Reject: I recommend rejecting the paper, but will not argue for my recommendation if the majority of other reviewers have a different opinion.

**Review:**

I appreciate the paper tackles a realistic problem and proposes a reasonable solution. The paper is well-motivated; however, there are some concerns I have with clarity as well as the experimental results.

Regarding clarity, I found there to be many instances in which I did not adequately understand what was being communicated. Generally, the paper would be much improved by taking care to explain things that are being introduced and with proper justification. I've listed some examples below:
* I strongly suggest summarizing the overall algorithm using pseudocode/algorithm box for clarity
* Figure 1 is very confusing without more context, so I would suggest either making this simpler or adding explanation
* Related work could use some more detail and additional explanation as to how these works relate to DROID specifically. For example, in addition to: "both fail to plan under missing/occluded terrain maps", an explanation of how the work addresses this problem would help.
* Section 5.3 does not contain enough information to fully understand the improvements, nor do the experimental results refer to these improvements

Regarding experiments, I find that the current results do not directly address all the claims made. First, I am not sure what each of the comparisons is *exactly* referring to---please consider making this a lot more explicit and explaining which of the claims each one is meant to test. Are all comparisons benefiting from the improvements? Otherwise, it's hard to understand what is providing the benefit---please make this more clear. Regarding the evaluation metrics, it's hard to judge whether the results are significant without justification for the metrics chosen. The most important issue, however, is that due to the lack of clarity (perhaps) I am not sure what to take away from the experiments and how they indeed validate the claims. For example, since the claim is that doing both is better in DROID, what is the effect of reward distillation only versus policy distillation only? It's currently unclear whether this is exactly being tested since the descriptions of the comparisons are a bit vague. Also, if the authors talk about improvements, it would be nice to include experiments that show their effects/analyze them.


**Quality Of The Limitations Section:**

Additional details required

**Questions For Rebuttal:**

* Can you benchmark/ablate on a couple other standard domains/tasks other than cartpole?
* How is $M$ chosen? is it assumed to be known? Relatedly, do you assume the data for different strategies are labeled?
* Section 5.1 says "We propose a novel reward distillation approach", but I was under the impression this was proposed in MSRD. Could you clarify exactly what MSRD did and how this component relates to it?
* Will the dataset/simulator be made publicly available?
* Can you show that the metrics chosen are correlated with some quantitative or qualitative measure that indicates task success (like Final Distance in the MPP)?



**Robotics Focus:**

Highly relevant to robotics but no hardware experiments

**Summary Of Paper:**

The paper proposes a new OLfD approach that addresses key problems in realistic settings where demonstrations are heterogeneous and limited. DROID modifies the OLfD method AVRIL by separating demonstrator/strategy-specific components and shared task components when inferring the reward function and learning an ensemble of policies that express these different strategies. The authors also introduce a new task/dataset comprised of paths by rover planners for a Martian Path Planning problem.

**Summary Of Recommendation:**

The paper proposes an impactful problem as well as a solution, but the clarity and some experimental analysis should be improved before publication.

---

### Decision · Program_Chairs · 2023-08-30

**Decision:**

Accept (Poster)

**Comment:**

The paper introduces a new approach for Learning from Demonstration in the offline setting with heterogeneous and limited demonstrations. Most reviewers agreed that the approach is interesting and novel, and appreciated the new realistic dataset which the authors introduced for their evaluation. They were also convinced by the experimental evaluation, especially after the additional ablations and comparisons in the thorough rebuttal. Please make sure to include them in the final version of the paper.